# The Effect of Toothpastes Containing Natural Extracts on Bacterial Species of a Microcosm Biofilm and on Enamel Caries Development

**DOI:** 10.3390/antibiotics11030414

**Published:** 2022-03-19

**Authors:** Aline Silva Braga, Mohamed Mostafa Hefny Abdelbary, Rafaela Ricci Kim, Fernanda Pereira de Souza Rosa de Melo, Luiz Leonardo Saldanha, Anne Lígia Dokkedal, Georg Conrads, Marcella Esteves-Oliveira, Ana Carolina Magalhães

**Affiliations:** 1Department of Biological Sciences, Bauru School of Dentistry, University of São Paulo, Bauru 17012-191, Brazil; aline_s.braga@hotmail.com (A.S.B.); rafaela.kim@usp.br (R.R.K.); 2Division of Oral Microbiology and Immunology, Department of Operative and Preventive Dentistry and Periodontology, RWTH Aachen University Hospital, 52074 Aachen, Germany; mabdelbary@ukaachen.de (M.M.H.A.); gconrads@ukaachen.de (G.C.); 3Department of Biological Sciences, School of Science, The São Paulo State University (UNESP), Bauru 17033-360, Brazil; fernandamelo405@gmail.com (F.P.d.S.R.d.M.); lluizsaldanha@gmail.com (L.L.S.); anne.dokkedal@unesp.br (A.L.D.); 4Department of Restorative, Preventive and Pediatric Dentistry, University of Bern, 3010 Bern, Switzerland

**Keywords:** antimicrobial strategies, dental caries, oral biofilms, plants

## Abstract

This study investigated the effects of herbal toothpaste on bacterial counts and enamel demineralization. Thirty-six bovine enamel samples were exposed to a microcosm biofilm using human saliva and McBain saliva (0.2% sucrose) for 5 days at 37 °C and first incubated anaerobically, then aerobically–capnophilically. The following experimental toothpaste slurries (2 × 2 min/day) were applied: (1) *Vochysia tucanorum* (10 mg/g); (2) *Myrcia bella* (5 mg/g); (3) *Matricaria chamomilla* (80 mg/g); (4) Myrrha and propolis toothpaste (commercial); (5) fluoride (F) and triclosan (1450 ppm F), 0.3% triclosan and sorbitol (Colgate^®^, positive control); (6) placebo (negative control). The pH of the medium was measured, bacteria were analyzed using quantitative polymerase chain reaction, and enamel demineralization was quantified using transverse microradiography. The total bacterial count was reduced by toothpaste containing *Myrcia bella*, *Matricaria chamomilla*, fluoride, and triclosan (commercial) compared to the placebo. As far as assessable, *Myrcia bella*, *Matricaria chamomilla*, and Myrrha and propolis (commercial) inhibited the outgrowth of *S. mutans*, while *Lactobacillus* spp. were reduced/eliminated by all toothpastes except *Vochysia tucanorum*. Mineral loss and lesion depth were significantly reduced by all toothpastes (total: 1423.6 ± 115.2 vol% × μm; 57.3 ± 9.8 μm) compared to the placebo (2420.0 ± 626.0 vol% × μm; 108.9 ± 21.17 μm). Herbal toothpastes were able to reduce enamel demineralization.

## 1. Introduction

The supragingival biofilm is a complex layer containing different microbial species and a rich extracellular matrix. The presence of sugars favors dysbiosis towards the acid-producing/acid-tolerant phenotype, which is the causative agent for caries development [1].

Mechanical disorganization of the biofilm via brushing with fluoride toothpaste is a solid strategy for controlling the disease [2]. A holistic approach could include mechanical removal of biofilms and a low-sugar diet, as well as microbial growth control to prevent dysbiosis. A caries-controlling strategy like is currently supported by the recent understanding of caries etiology and widely accepted ecological plaque hypothesis [3]. Accordingly, medicinal plants have long been studied because they are a rich source of polyphenols, terpenoids, and alkaloid compounds with antimicrobial, anti-biofilm, anti-glycosyltransferase (dextran reduction), and anti-caries activities. Furthermore, phytotherapy is known as a new approach for treating and/or preventing diseases, which may induce lower side effects and have lower costs compared to traditional therapies [4,5].

Brazil has one of the greatest floral diversities worldwide and encompasses two biodiversity hotspots [6] for conservation and exploitation priorities: the Brazilian savanna (Cerrado) and Atlantic rainforest (Mata Atlântica). *Myrcia bella* Cambess. (Myrtaceae) and *Vochysia tucanorum* Mart. (Vochysiaceae) are species that occur naturally in the Cerrado and have medicinal potential [7,8,9].

On the other hand, *Matricaria chamomilla* L. (Asteraceae) is a well-known medicinal plant used in traditional medicine for oral treatments in Europe and Western Asia, and in contrast to the above-mentioned plants, it has already been included in oral care products. This plant has been tested as an antimicrobial agent against some bacterial species [10,11], and is effective against dental caries when used as a solution [12,13]. Recently, experimental solutions containing 5 mg/mL *Myrcia bella*, 10 mg/mL *Vochysia tucanorum*, or 80 mg/mL *Matricaria chamomilla* were evaluated in a microcosm biofilm model. Only the *Matricaria chamomilla* solution was able to significantly reduce enamel demineralization, similar to a commercial solution containing *Malva sylvestris* [13].

Considering the above results [13], the inclusion of natural extracts into toothpastes could combine their chemical action on biofilm and enamel with the mechanical removal of dental biofilm by brushing. As well as active agents, toothpastes have other ingredients, such as detergents, that could improve the effectiveness of the herbals. Furthermore, it is expected that better compliance of the patient can be achieved in the daily oral hygiene routine when he/she has to apply only one product, for example, brushing with toothpaste containing natural extracts, rather than rinsing with a solution containing a natural extract after brushing with regular toothpaste.

Therefore, the present study aimed to investigate and compare the anti-caries effects of different experimental toothpastes containing natural extracts (*V. tucanorum, M. bella,* and *M. chamomilla*), already tested as a solution, compared to both commercial toothpastes and a placebo toothpaste. Their effect on the total microorganism level and enamel demineralization was evaluated using quantitative polymerase chain reaction (qPCR) and transverse microradiography (TMR).

## 2. Material and Methods

### 2.1. Ethical Aspects and Saliva Collection

The Ethics Committee (CEEA 84325518.2.0000.5417) of the Bauru School of Dentistry (Bauru, Brazil) approved the study. Ten healthy participants (23.8 ± 3 years old, eight women and two men, with signed informed consent) participated in the study as saliva donors. The definition of the inclusion criteria for saliva donors as well as the procedures for saliva collection under stimulation (salivary flow > 1 mL/min, 10 min of collection) followed previously reported protocols [13,14]. Saliva samples from 10 donors were mixed to get a saliva pool that was applied to standardize the biofilm cultivation. The saliva pool was diluted in glycerol (70% saliva and 30% glycerol) and 1 mL aliquots were stored at −80 °C [15].

### 2.2. Extracts and Toothpaste Compositions

*Vochysia tucanorum* Mart., and *Myrcia bella* Cambess. leaf samples were collected at the Jardim Botânico Municipal de Bauru (Bauru, Brazil), (22°20′41.4″ S-49°01′45.1″ W). Exsiccates were deposited in the Herbarium of UNESP (UNBA) under code numbers HRCB59831 and UNBA6034. The collections have authorization issued by SISBIO (System for the control of biological material in research) under code number 39825-1 and CGEN number 010468/2014-51. Fresh leaves were hot air dried at 45 °C and ground in a knife mill. The powdered leaves were extracted with EtOH: H_2_O 7:3% (*v/v*) via percolation at room temperature [16,17]. The filtrates were concentrated under dryness and reduced pressure at 40 °C. They were finally lyophilized, yielding the hydroalcoholic leaves’ extract (70% EtOH) of *V. tucanorum* (17% dry weight) and *M. bella* (28% dry weight) [13]. *Matricaria chamomilla* L. (flower and stalk) dry extract was purchased from Quimer Insumos Vegetais (São Paulo, Brazil).

Table 1 shows the compositions of the experimental and commercial toothpastes. The concentrations of the natural extracts were determined according to the results of minimum bactericidal concentration (MBC) previously tested on *S. mutans* strain ATCC25175 as a reference under aerobic conditions (37 °C, 5% CO_2_) [12]. Since the MBC values, reflecting the antimicrobial activity of the natural extracts, were tested in a range (it was applied as a serial dilution from 20 to 0.03907 mg/mL for *Vochysia tucanorum* and *Myrcia bella*; and from 160 to 2.50 mg/mL for *Matricaria chamomilla)*, their concentrations applied in toothpastes were also different. The pH of the toothpaste samples varied from 7.0–8.7.

### 2.3. Tooth Specimen Preparation

Bovine teeth were donated by cattle slaughtered in the food manufacturing industry (Frigol S. A., Lençóis Paulista-SP, Brazil) after approval by the Ethics Committee on Animal Use (CEUA, Number: 002/2018, Bauru School of Dentistry, University of São Paulo, Bauru, Brazil). Forty bovine enamel specimens (4 mm × 4 mm) were prepared and thirty-six were selected based on the average roughness (Ra) (contact profilometer Mahr, Göttingen, Germany) [13] to standardize the enamel surface for biofilm growth. Two parts of each 1/3 of the enamel surface were covered with red nail polish (Estreia-Colorama, Rio de Janeiro, Brazil) to create two reference areas (sound enamel), enabling an appropriate analysis of enamel demineralization by TMR. Thereafter, the specimens were sterilized using ethylene oxide (gas exposure time (30% ETO/70%CO_2_) for 4 h under 0.5 ± 0.1 kgF/cm^2^ pressure). Enamel specimens were randomly distributed into six groups (n = 6, Table 1) using their mean Ra (Ra: 0.164 ± 0.03 µm) as criteria.

### 2.4. Microcosm Biofilm Formation and Treatments

Microcosm biofilms were created in biological duplicates, with n = 3 in each experiment (n = 6). The human saliva pool (inoculum) was mixed with McBain saliva [18] prepared as described by Braga et al. [13]. Microcosm biofilms were produced on enamel under anaerobiosis for three days (to allow growth of anaerobic oral species) and changed to aerobiosis (7% CO_2_) for the last two days, to allow the presence of facultative species at the biofilm surface.

Each enamel specimen was fixed in a 24-well microtiter plate by using liquid silicon at the bottom of each well. Human saliva and McBain saliva solution were added (v = 1.5 mL/well) and the plates were incubated anaerobically in GasPak (BD, Sparks, NV, USA) at 37 °C for the first 8 h. Thereafter, the specimens were washed with PBS and exposed to fresh McBain saliva containing 0.2% sucrose and incubated under the same conditions until completing the first day of the experiment.

From the second to the fifth day, the specimens were treated with the experimental toothpaste slurries (1 g of each toothpaste in 3 mL of deionized water) (Table 1) at room temperature, twice a day (1 mL/well, 2 min). The specimens were washed using PBS (twice) and fresh McBain saliva containing 0.2% sucrose was added. Then, the microplates were incubated under strictly anaerobic conditions at 37 °C for additional two days, and for the last two days, under 7% CO_2_ and 37 °C [13,19]. Figure 1 summarizes the biofilm model, and Figure 2 shows the 24-well microtiter plates during and after the experiment, in which it is possible to see biofilm on the bottom of the wells.

### 2.5. pH Monitoring

The pH of the medium (McBain artificial saliva) in the layer immediately above the biofilm was monitored after the first 8 h and 24 h of biofilm formation. On days 2, 4, and 5, the medium pH was measured before performing the treatments using a MiniTrode (Hamilton, Bonaduz, Switzerland).

### 2.6. Biofilm Analysis: Molecular Analyses (Genome Count Determination)—qPCR

After five days of biofilm formation, the specimens were removed from the 24-well-plates (leaving a characteristic imprint as proof of biofilm formation, Figure 2) and transferred into Eppendorf tubes containing 250 µL of 0.9% saline solution with three glass spheres (3 mm in diameter), and stored in a freezer at −70 °C. The next day, the microtubes were defrosted and specimens were vortexed for 10 s to provoke complete removal of the biofilm from the specimen surface [20]. This suspension was transferred to a new microtube. For washing, a volume of 250 µL sterile bidistilled water was added to the biofilm pellets, vortexed, and cells were again pelleted by centrifugation at 10,600 rcf for 2 min, followed by the subsequent discarding of the supernatant. After the addition of 250 µL sterile bidistilled water and suspension, an aliquot of 100 µL was used for DNA extraction using the QIAamp DNA Mini Kit (QIAGEN, Düsseldorf, Germany) according to the manufacturer’s protocol. The DNA samples were stored at −20 °C.

Bacterial numbers were analyzed by qPCR using a *QuantStudio* cycler (ver. 3; Applied Biosystems, Thermo Fisher Scientific, Waltham, MA, USA), as previously described by Henne et al. [20,21]. Specific primers were used to measure total bacteria, *Streptococcus mutans*, and *Lactobacillus* spp. under conditions previously published by our group [13].

Approximately 0.1 µL Primer Forward, 0.1 µL Primer Reverse, 10 µL of Rastem/Master Mix solution, and 8.8 µL water were pipetted into 96-well plates containing the respective primers. Approximately 0.1 µL of DNA samples were also pipetted into each well. For the calibration curve, known concentrations of the tested bacteria were applied (positive control) and water was used as a negative control. The data of biological triplicates were obtained using QuantStudio Design & Analysis software v.1.4.3 and exported to Microsoft Excel [22,23].

### 2.7. Demineralization Analysis: TMR

The enamel specimens were cleaned, transversally sectioned, polished, and subjected to TMR (20 kV and 20 mA, Softex, Tokyo, Japan). The developed plate was analyzed using a transmitted light microscope fitted with a 20× objective. Two images per specimen (250 μm × 400 μm) were obtained using data acquisition and interpreted using calculation software from the Inspector Research System (Amsterdam, The Netherlands). The mineral content was calculated assuming 87 vol% of mineral content for sound enamel and that the lesion depth ended when the enamel contained approximately 82.5% of mineral volume. The integrated mineral loss (ΔZ, vol% × μm) and lesion depth (LD, μm) were calculated from two images per specimen.

### 2.8. Statistical Analysis

Data were statistically compared using *GraphPad Instat* and *GraphPad Prism* software (*GraphPad* Software version 8, San Diego, CA, USA). Distribution and homogeneity were tested using Kolmogorov–Smirnov’s and Bartlett’s tests, respectively. With respect to the medium pH, the data were compared using two-way ANOVA/Tukey’s test (factors: treatment and periods of analysis: 8 h, 24 h, 72 h, 96 h, and 120 h) and mixed model analysis with dummy variables (Software SPSS Statistics version 28.0). Bacterial levels were analyzed using Kruskal–Wallis and Dunn′s multiple comparisons test, while the TMR comparison was performed using ANOVA and Tukey’s test. The level of significance was set at 5%.

## 3. Results

### 3.1. pH Changes

Table 2 shows the mean pH values during the microcosm biofilm growth. Switching from sucrose-free (over 8 h) to sucrose-containing (0.2%) medium (over 16 h) resulted in a significant drop in pH by one unit. After 72 h, the pH values significantly increased compared with those at 24 h and remained constant until the end of the experiment at t = 120 h. All toothpastes containing natural extracts resulted in medium pH values (measured in the medium directly above the biofilm) that were higher than the pH value of the placebo toothpaste (negative control), while the pH of the medium treated with F and triclosan toothpaste (commercial) did not differ significantly from those of the placebo or also from the experimental toothpastes (2-way ANOVA, *p* < 0.05).

Mixed model analysis showed a significant effect of time (dummy coded with 8 h without sucrose as the reference, *p* = 0.000 for all periods: 24, 72, 96, and 120 h) and among the treatments (dummy coded with placebo as the reference group, *p* = 0.000 for all treatments). In the interaction analysis, *Matricaria chamomilla* L. (*p* = 0.059) and F and triclosan toothpaste (*p* = 0.118) toothpastes did not differ from the placebo at 24 h. However, at the other periods, and for the other treatments in all periods, significant differences were found compared to the placebo.

### 3.2. qPCR and Changes in Numbers of Bacteria

Figure 3 shows the genome counts for total bacteria (Figure 3a), *Lactobacillus* spp. (Figure 3b), and *Streptococcus mutans* (Figure 3c) recovered from the microcosm biofilm formed on the enamel specimens. A significant reduction in total bacteria was caused by the experimental *Myrcia bella* Cambess. and *Matricaria chamomilla* L. toothpastes, as well as by the F and triclosan toothpastes, while all the other toothpastes were similar to the placebo (negative control) (Kruskal–Wallis, *p* < 0.0001, Figure 3a).

Complete elimination (values under the detection limit) or inhibition of out-growth of *Lactobacillus* spp. was caused by *Matricaria chamomilla*, and by both commercial toothpastes, Myrrha and propolis toothpaste, and F and triclosan toothpaste (Kruskal–Wallis, *p* = 0.0005). The level of inhibition of *Lactobacillus* spp. from *Myrcia bella* was significantly higher than that of the placebo (*p* = 0.003), while *Vochysia tucanorum* was not significantly different from that of the placebo (negative control) (Figure 3b).

The experimental *Myrcia bella*, *Matricaria chamomilla*, and commercial Myrrha and propolis toothpastes completely reduced *S. mutans* under the detection limit, whereas this species was observed in biofilms belonging to *Vochysia tucanorum*. Among the toothpastes containing F and triclosan, *Vochysia tucanorum*, and the placebo (negative control), there was no significant difference (Kruskal–Wallis, *p* > 0.99, Figure 3c). However, some tests (including the placebo) were negative for *S. mutans*, thus, weakening their overall significance.

### 3.3. TMR and Enamel Demineralization

Integrated mineral loss and lesion depth were reduced by all toothpastes that contained active agents (mineral loss approximately 42% and lesion depth approximately 47% lower) when compared to the placebo (negative control) (ANOVA, *p* = 0.0011 and *p* < 0.0001, respectively). Figure 4 and Table 3 show the results and lesion profiles of the representative specimens from each group.

## 4. Discussion

The number of marketed oral care products with natural active ingredients is steadily increasing, but there is little scientific evidence regarding their oral health benefits and their potential to reduce dental caries development [24]. In this context, the anti-caries efficiency of some natural extracts was successfully tested in a previous study where they were incorporated into mouth rinses [13]. However, as the delivery of preventive agents by means of toothpastes has been, up to now, the most successful way to reduce caries prevalence worldwide, it would be of great importance to also investigate their effectivity in this kind of formulation. Therefore, in the present study, the anti-caries efficiency of similar natural plant extracts [13] was tested and incorporated into experimental toothpastes which have the potential to substantially improve the clinical relevance of this preventive strategy.

The microcosm biofilm model applied in our study has the advantage of retaining most of the cultivable microorganisms found in the oral cavity [25]. We used a protocol to allow the growth of both obligate and facultative anaerobes out of the donor saliva and to reproduce a natural biofilm surface that matures under sugar consumption. The biofilm growth was visually controlled daily. Additionally, daily measurements of the biofilm pH were performed since it has been demonstrated in previous studies that there is a strong correlation between the changes in the biofilm pH (especially in the presence of sucrose) and the changes in the relative bacterial/metabolic activity of the biofilm of natural cavitated caries lesions (in vivo) [20,24] and the ribosome counts of caries-associated species in co-cultures in vitro [22]. 

The commercial Myrrha- and propolis-containing toothpaste and the fluoride and triclosan-containing toothpaste were included in the present study because the former had herbal ingredients and the latter is a regular toothpaste that has previously been shown to have both antimicrobial and anti-caries effects [14]. Therefore, both toothpastes were used as the positive controls. A previous study has shown the antimicrobial effect of *Myrrha* and propolis toothpaste using a viability assay on aerobic microcosm biofilm [14], in agreement with our results. However, the anti-caries effect (reduction of demineralization) was lower in the cited study [14] compared to the present one, which might be due to differences in the microcosm biofilm model that was completely aerobic, and the toothpaste was applied only once a day for 60 s in the former study [14]. One limitation of our study is that we have no further information on the exact extraction process of the herbals contained in the commercial toothpastes (not provided by the manufacturer). Therefore, the results here should be taken with caution. For example, the chemical composition of propolis depends on the area where the insects collected the material, seasonality, and the flowering plants there. A previous study showed that several of the compounds identified in propolis are able to inhibit glycosyltransferase—GTF activities and bacterial growth. Apigenin, for example, is a potent inhibitor of GTF activity, and tt-farnesol was found to be an effective antibacterial agent [26]. On the other hand, there is no information in the literature about a possible mechanism of action of Myrrha on oral bacteria.

With respect to the experimental natural extracts, a previous investigation [13] found no antimicrobial effect when they were used in the same concentration and in a similar microcosm biofilm model used here but applied as solutions. In the present study, toothpastes containing *M. bella* and *M. chamomilla* reduced the total number of bacteria, *Lactobacillus* spp. and *S. mutans*. The effect can be explained by the presence of the active compound but also, at least partially, by the interaction of different substances from the toothpaste that enhance the action of the natural agent, especially increasing the pH values above the biofilm. The high pH impairs the growth of aciduric bacteria as *Lactobacillus* spp. and *S. mutans* [27,28]. However, *V. tucanorum*, even when formulated as a toothpaste, was not able to reduce *Lactobacillus* spp. and *S. mutans*.

The hydroalcoholic extract of *M. bella* leaves has already been described in the literature to exhibit antimicrobial properties against *Escherichia coli* [29]. A previous chemical study has identified several phenolic compounds as flavonoid *O*-glycosides, phenolic acids, and hydrolyzable tannins derivatives of gallic and ellagic acid in *M. bella* leaves [30,31]. Studies in the literature describe the antimicrobial activity of flavonoid derivatives of quercetin and myricetin [32] and hydrolysable tannins [33]. Some of the proposed mechanisms of the antibacterial action for tannins are the inhibition of extracellular microbial enzymes and oxidative phosphorylation [34] and the disruption of cell membrane permeability [35,36]. The mechanism of some flavonoids was attributed to the inhibition of DNA gyrase and cytoplasmatic membrane functions [32].

The hydroalcoholic extract of *V. tucanorum* leaves contains pentacyclic oleanane triterpenoids and flavonoid derivatives as main constituents [8,37]. To the best of our knowledge, no data about antimicrobial activity for this extract is available in the literature.

*Matricaria chamomilla* is effective against gram-positive and gram-negative bacteria [38]. Chamomile flowers contain more than 120 chemical constituents, including sesquiterpenes, coumarins, flavonoids, and polyacetylenes [39]. Existing evidence suggests that the antimicrobial effects of chamomile may be attributable to its terpenic derivatives, chamazulene, β-bisabolol, and A and B bisabolol oxides [40]. The mechanism of action of terpenes is not fully understood but is speculated to involve the membrane disruption of bacteria by the lipophilic compounds [41].

The anti-caries effect of the experimental toothpaste was similar to that of a commercial toothpaste containing fluoride and triclosan (Colgate^®^ Total 12), whose action is already well-known [42]. Triclosan is an antibacterial agent that affects bacterial growth by inhibiting key bacterial metabolic pathways. This action may be responsible for reducing the total bacterial and *L. paracasei* load in the biofilm when fluoride toothpaste containing triclosan was compared to a conventional fluoride toothpaste [42,43]. In contrast, fluoride and triclosan toothpaste did not reduce the level of *S. mutans* in the present model, which deserves further study, especially because *S. mutans* did not grow appropriately. This toothpaste has other important active ingredients besides triclosan - fluoride - which acts directly on the enamel surface, reducing enamel demineralization and increasing enamel remineralization, which justified the reduced caries lesion formation [44].

Although most toothpastes, except *V. tucanorum,* had antimicrobial effects on the analyzed species, all experimental toothpastes were able to increase the biofilm pH and reduce enamel demineralization compared to the placebo. The effect on the biofilm pH may not only be due to the active components but also their combination with common ingredients of the toothpastes that could interact with the biofilm matrix, thus, enhancing its pH. The high pH value found in the biofilm (from 48 to 120 h), rather than the antimicrobial action itself, as discussed above, might have been responsible for the low demineralization presented by the enamel specimens treated with toothpastes. It could also be responsible for the placebo controls devoid of *S. mutans*, as this species survives in an acidic environment only [27].

The new strategies must act by inhibiting cariogenic virulence, modulating rather than eliminating the microbiome of dental biofilm, and preferably, with low side effects [4]. The most important outcome is not related to the effect on biofilm alone, but it is related to the reduction of tooth demineralization. Our study provides new insights into the effect of the experimental toothpastes in reducing enamel demineralization (caries development), which may be due to some ingredients of the toothpastes that could have some additive effects. As an example of some toothpaste ingredients that could be important in this context, carboxymethyl cellulose, an anionic polysaccharide present in toothpastes and obtained by the alkali-catalyzed reaction of cellulose with chloroacetic acid, has been studied in different edible film formulations because of its film-forming ability, biodegradability, biocompatibility, and colorless, tasteless, toxicologically harmless, inexpensive, and hydrophilic nature. However, it does not possess any intrinsic antimicrobial activities [45], although when the essential oils of *Santolina chamaecyparissus, Schinus molle,* and *Eucalyptus globulus* are incorporated in carboxymethyl cellulose films, they have significant antimicrobial properties [46]. Sodium lauryl sulfate, a detergent, has been shown to exert antimicrobial effects against *S. mutans* [47]. Therefore, we cannot exclude their role in the protective effects of toothpastes.

In conclusion, all tested toothpastes were able to reduce the formation of enamel caries in this experimental model, despite only some toothpastes showing antimicrobial action on the analyzed species. However, these results have still to be confirmed in further studies using other more clinically similar models, such as in situ and in vivo models, and by including other analyses, such as the analyses of the microbiome/metabolome before clinical use can be recommended. Furthermore, there is still a need for tests with fractions and isolated compounds to identify the antimicrobial active principles towards a better understanding of the mechanisms involved in their anti-caries action on microcosm biofilms.

## Figures and Tables

**Figure 1 antibiotics-11-00414-f001:**
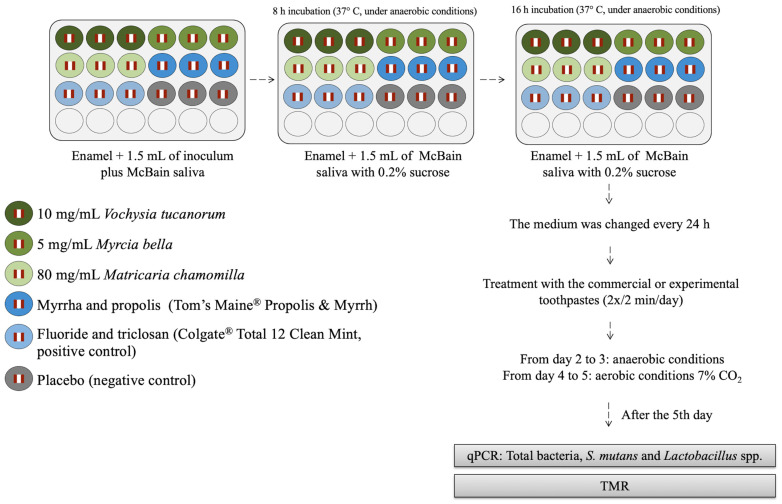
Experimental microcosm biofilm protocol as well as the correspondent treatments and the response variables applied to analyze the biofilm and the tooth.

**Figure 2 antibiotics-11-00414-f002:**
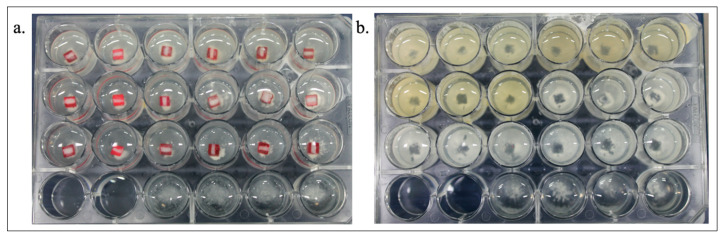
The image shows the 24-well microtiter plates during the experiment (**a**) and after the removal of enamel specimens, leaving a print in the biofilm (**b**).

**Figure 3 antibiotics-11-00414-f003:**
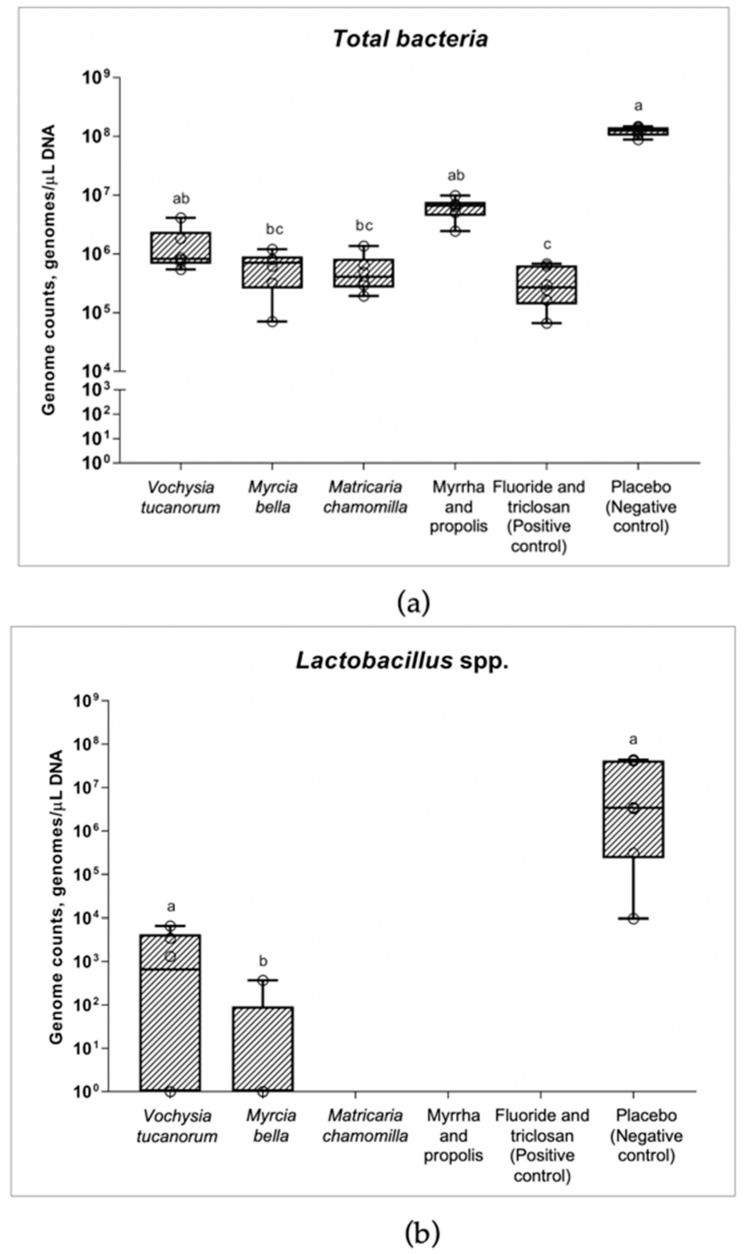
Boxplot of genome counts of total bacteria, *Lactobacillus* spp., and *Streptococcus mutans* present in microcosm biofilm after treatments. (**a**) Boxplot of genome counts of total bacteria present in microcosm biofilm after treatments (Kruskal–Wallis/ Dunn’ multiple comparison test, *p* < 0.0001). (**b**) Boxplot of genome counts of *Lactobacillus* spp. present in microcosm biofilm after treatments (Kruskal–Wallis/ Dunn’ multiple comparison test, *p* = 0.003). *Lactobacillus* spp. (n = 3 *Vochysia tucanorum*, n = 1 *Myrchia bella*, n = 6 placebo). (**c**) Boxplot of genome counts of *Streptococcus mutans* present in microcosm biofilm after treatments (Kruskal–Wallis/ Dunn’ multiple comparison test, *p* > 0.99). *S. mutans* (n = 2 *Vochysia tucanorum*, n = 1 F and triclosan toothpaste, n = 1 placebo). Genomes per microliter of samples were determined by qPCR. Different letters show significant differences among the treatments.

**Figure 4 antibiotics-11-00414-f004:**
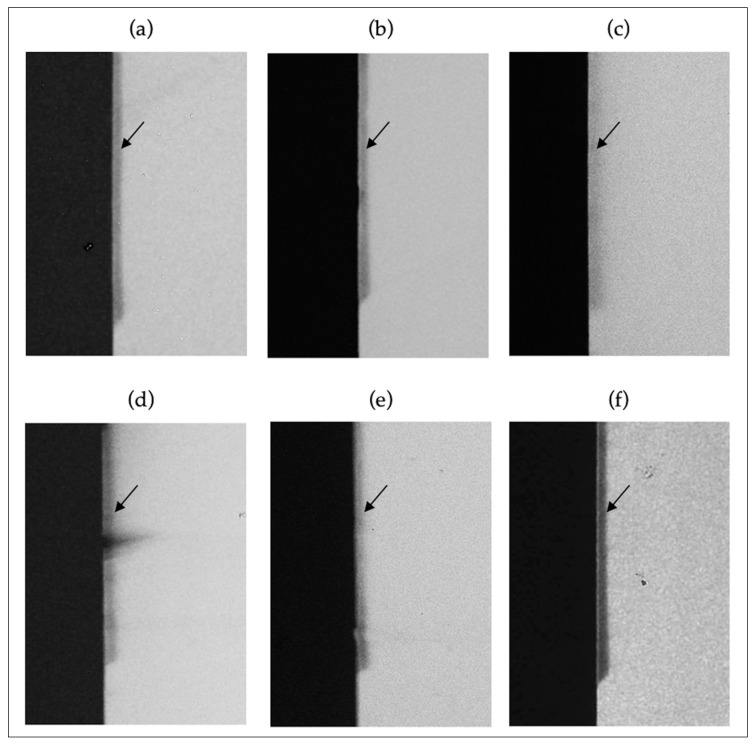
Representative TMR pictures (20×) of the artificial carious enamel lesions created using microcosm biofilm after treatment with different toothpastes. (**a**) *Vochysia tucanorum;* (**b**) *Myrcia bella;* (**c**) *Matricaria chamomilla;* (**d**) Myrrha and propolis toothpaste; (**e**) F and triclosan toothpaste (positive control)*;* (**f**) Placebo (negative Control). The black arrow shows the initial caries lesions (darker in color). Placebo showed a more radiolucent area (darker) indicating more enamel demineralization compared to the other groups, which were more radiopaque (brighter), thus, less demineralized.

**Table 1 antibiotics-11-00414-t001:** Experimental and commercial toothpastes.

Experimental and Commercial Toothpastes	Company/City, Country	Composition/Extract Concentration
*Vochysia tucanorum*	Pharmacy Specifíca (Bauru/São Paulo, Brazil)	Carboxymethylcellulose, glycerin, sodium methyl p-hydroxybenzoate, saccharin, hydrated silica, titanium dioxide, sodium lauryl sulfate, water. Active component: *Vochysia tucanorum* (10 mg/g, 70% EtOH leaf extract).
*Myrcia bella*	Pharmacy Specifíca (Bauru/São Paulo, Brazil)	Carboxymethylcellulose, glycerin, sodium methyl p-hydroxybenzoate, saccharin, hydrated silica, titanium dioxide, sodium lauryl sulfate, water. Active component: *Myrcia bella* (5 mg/g, 70% EtOH leaf extract).
*Matricaria chamomilla*	Pharmacy Specifíca (Bauru/São Paulo, Brazil)	Carboxymethylcellulose, glycerin, sodium methyl p-hydroxybenzoate, saccharin, hydrated silica, titanium dioxide, sodium lauryl sulfate, water. Active component: *Matricaria Chamomilla* (80 mg/g, commercial aqueous flower and stalk extract).
*Commiphora myrrha* and propolis (Tom’s Maine^®^ Propolis & Myrrh)—Myrrha and propolis toothpaste	Tom’s Maine^®^/Kennebunk, USA	Calcium carbonate, glycerin, water, hydrated silica, xylitol, sodium lauryl sulfate, xantan gum, benzyl alcohol, natural favor, *Commiphora myrrha* resin extract (myrrh), propolis extract. Active component: xylitol, *Commiphora myrrha* resin extract (myrrh), propolis extract.
Sodium Fluoride + 0.3% triclosan and sorbitol (Colgate^®^ Total 12 Clean Mint, positive control)—Fluoride and triclosan toothpaste	Colgate-Palmolive/São Paulo, Brazil	Hydrated silica, sodium lauryl sulfate, PVM/ MA, copolymer, flavor, carrageenan, sodium hydroxide, sodium saccharin, titanium dioxide (CI 77891), dipentene, water. Active component: Sodium fluoride (1450 ppm F), 0.3% triclosan and sorbitol.
Placebo (Negative control)	Pharmacy Specifíca (Bauru/São Paulo, Brazil)	Carboxymethylcellulose, glycerin, sodium methyl p-hydroxybenzoate, saccharin, hydrated silica, titanium dioxide, sodium lauryl sulfate, water. Active component: none.

**Table 2 antibiotics-11-00414-t002:** The medium pH values during the microcosm biofilm growth.

Treatment	pH Values (Mean ± SD)				
	8 h ^B^	24 h ^A^	72 h ^C^	96 h ^C^	120 h ^C^
*Vochysia tucanorum* ^a^	5.51 ± 0.02	4.27 ± 0.04	6.60 ± 0.07	6.82 ± 0.09	6.92 ± 0.10
*Myrcia bella* ^a^	5.52 ± 0.01	4.23 ± 0.02	6.57 ± 0.03	6.79 ± 0.04	7.04 ± 0.01
*Matricaria chamomilla* ^a^	5.50 ± 0.01	4.22 ± 0.03 *	6.60 ± 0.02	6.76 ± 0.10	6.98 ± 0.06
Myrrha and propolis ^a^	5.51 ± 0.03	4.24 ± 0.03	6.50 ± 0.06	6.55 ± 0.54	6.82 ± 0.10
Fluoride + triclosan (positive control) ^ab^	5.52 ± 0.05	4.22 ± 0.01 *	6.03 ± 0.21	5.89 ± 0.19	6.08 ± 0.04
Placebo (negative control) ^b^	5.55 ± 0.02	4.18 ± 0.01 *	5.03 ± 0.07	4.74 ± 0.09	5.88 ± 0.02

Different lowercase letters (at rows) show significant differences among the treatments considering all times of biofilm formation. Different uppercase letters (at columns) show significant differences among the times of biofilm formation for all treatments. Two-way ANOVA/Tukey’s multiple comparison test (treatment *p* < 0.0001; time *p* = 0.0033). Mixed model analysis showed significant effect of time (dummy coded with 8 h- without sucrose and without treatment as the reference, with *p* = 0.000 for all periods: 24 h with sucrose but without treatment, and 72, 96, and 120 h with both sucrose and treatment) and among the treatments (dummy coded with Placebo as the reference group, with *p* = 0.000 for all treatments), * except *Matricaria chamomilla* (*p* = 0.059) and F and triclosan toothpaste (*p* = 0.118) toothpastes that did not differ from placebo at 24 h.

**Table 3 antibiotics-11-00414-t003:** Mean ± SD of the integrated mineral loss (ΔZ, vol% × μm) and the lesion depth (LD, μm) of enamel.

Treatment	ΔZ (vol% × μm)	LD (μm)
*Vochysia tucanorum*	1515.0 ± 358.1 ^a^	57.1 ± 14.0 ^a^
*Myrcia bella*	1460.0 ± 345.7 ^a^	52.7 ± 24.5 ^a^
*Matricaria chamomilla*	1540.0 ± 191.5 ^a^	74.2 ± 10.5 ^a^
Myrrha and propolis	1293.3 ± 293.2 ^a^	53.5 ± 22.6 ^a^
Fluoride + triclosan (positive control)	1310.0 ± 277.4 ^a^	49.3 ± 11.4 ^a^
Placebo (negative control)	2420.0 ± 699.0 ^b^	108.9 ± 21.2 ^b^

Different letters in the same column show statistical differences between groups. All parameters were compared using ANOVA/Tukey (ΔZ: *p* = 0.0011; LD: *p* < 0.0001).

## Data Availability

Not applicable.

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
