# Peer review of "The Effect of Toothpastes Containing Natural Extracts on Bacterial Species of a Microcosm Biofilm and on Enamel Caries Development"

_antibiotics, 2022, doi:10.3390/antibiotics11030414_

Round 1
Reviewer 1 Report
The paper by Braga and colleagues describes a very interesting and innovative study focused on demonstrating the antimicrobial and anti-caries activities of natural extracts containing toothpaste.
Several minor points need to be addressed before publication:
Lines 90-92: A more detailed description of the extracts should be provided
Table 1: how the concentration of natural extracts to be added have been chosen?
Table 2 is quite hard to be read. pH values should be more distantiated
Figure 3 should be moved before table 3 In addition, the use of arrows or other symbols should be used for the description of TMR pictures
Author Response
Thank you for the review of our manuscript entitled “The effect of toothpastes containing natural extracts on bacterial species of a microcosm biofilm and on enamel caries development”. We appreciate all the suggestions and comments made about this manuscript. Please find below the answers for the reviewers’ comments. The changes were marked up using the “Track changes” in the main document.
Reviewer: 1
The paper by Braga and colleagues describes a very interesting and innovative study focused on demonstrating the antimicrobial and anti-caries activities of natural extracts containing toothpaste.
Several minor points need to be addressed before publication:
Lines 90-92: A more detailed description of the extracts should be provided
Table 1: how the concentration of natural extracts to be added have been chosen?
Table 2 is quite hard to be read. pH values should be more distantiated
Figure 3 should be moved before table 3 In addition, the use of arrows or other symbols should be used for the description of TMR pictures
Answer: Thank you for your comments and support. The detailed description of the process for obtaining the extracts was added in lines 91 to 107, as requested. The concentrations of natural extracts were chosen from previous results (MBC tests with Streptococcus mutans ATCC25175 strain as a reference). Since the extracts presented different MBC values, we tested toothpastes with different concentrations of the plants extracts. We revised Table 2 and Figure 3 following the comments.
Reviewer 2 Report
The present manuscript shows some evidence of anti-biofilm activity of some natural extracts containing toothpaste. The manuscript is well-written, but I have the following suggestions:
Major issues:
The manuscript only showed some anti-biofilm activities, there are no mechanistic studies. The manuscript is just a report, not a full research article.
The used natural components were of different concentrations (5 to 80 mg/ml, commercial unknown) which makes the study complicated, the effects of the components can’t be compared. Why were such different concentrations used?
There is no evidence for the initial biofilm formation on all of the tested enamel after the initial incubations. Data should be presented as before treatment, after treatment for better illustration of the study results. They should use 2 sets for this study.
There is no image of biofilms. Representative biofilm images would improve the manuscript much better.
Minor issues:
Table 2 is too crowded, difficult to distinguish between each value.
Figure 2 X-axis samples names are too small.
There are no mentions of the carious enamel lesions in Figure 3. Please mark them in the figures for a clear demonstration of the differences.
Author Response
Thank you for the review of our manuscript entitled “The effect of toothpastes containing natural extracts on bacterial species of a microcosm biofilm and on enamel caries development”. We appreciate all the suggestions and comments made about this manuscript. Please find below the answers for the reviewers’ comments. The changes were marked up using the “Track changes” in the main document.
Reviewer: 2
The present manuscript shows some evidence of anti-biofilm activity of some natural extracts containing toothpaste. The manuscript is well-written, but I have the following suggestions:
Major issues:
The manuscript only showed some anti-biofilm activities, there are no mechanistic studies. The manuscript is just a report, not a full research article.
The used natural components were of different concentrations (5 to 80 mg/ml, commercial unknown) which makes the study complicated, the effects of the components can’t be compared. Why were such different concentrations used?
Answer: Thank you for your comments and support. This reviewer is right, that we are presenting MBC results of complex natural extracts only. We regret that the antimicrobial mechanisms are not addressed in our paper but every discovery of (new) antimicrobial agents started by measuring the inhibitory effect. The different concentrations of natural extracts used were based on previous study (Braga AS, de Melo Simas LL, Pires JG, Souza BM, de Souza Rosa de Melo FP, Saldanha LL, et al. Antibiofilm and anti-caries effects of an experimental mouth rinse containing Matricaria chamomilla L. extract under microcosm biofilm on enamel. J Dent. 2020;99:103415). We decided to use the MBC results from Streptococcus mutans ATCC25175 strain as a reference. Since the extracts presented different MBC values reflecting their different antimicrobial activity, we tested toothpastes with different concentrations of the plants extracts.
There is no evidence for the initial biofilm formation on all of the tested enamel after the initial incubations. Data should be presented as before treatment, after treatment for better illustration of the study results. They should use 2 sets for this study.
Answer: The pH analyses were performed without treatment and sucrose only at 8h of biofilm formation. At 24h, the biofilm has been exposed to sucrose, but not to treatment yet. Thereafter, the pH analyses were done over biofilm exposed to sucrose and treatments. The molecular analysis of biofilm was done only at 5th day (the end of cultivation). The explanation has been provided in the Table 2. Moreover, in accordance with the modified ecological plaque hypothesis (Marsh, P. D. In sickness and in health-what does the oral microbiome mean to us? An ecological perspective. Adv Dent Res. 2018;29:60–5) on the formation of the caries lesions in enamel (Figure 3), is a clear surrogate of biofilm formation over samples, as otherwise, no demineralization could have occurred.
There is no image of biofilms. Representative biofilm images would improve the manuscript much better.
Answer: We are attaching two images for the attention of this reviewer just to show how obvious biofilm formation was in our McBain-saliva-sucrose microcosm model. The images are from a pre-experiment (placebo control) where we first optimized the conditions. The upper part shows the test specimens in wells covered with biofilm, the lower part the remaining biofilm after removing the specimens (a footprint is obvious formed by the broken biofilm) (Image A). Due to the fact that we never had a reason to question the formation of biofilm, we did not thought about including a biofilm picture. The other image shows the 24-well microtiter plate with specimens containing biofilm, during the experiment (Image B). We integrated image A into Figure 1. Thank you.
Reviewer 3 Report
Details regarding the herbal extracts: Although reference is made to an earlier publication the authors should specify the herbal preparations as precise as possible (kind and strength of the extraction solvent, drug-extract-ratio, type of the extract (liquid extract, dry extract). Without these details the statements regarding the content (e.g. Vochysia tucanorum 10 mg/g) are meaningless.
In table 1 the ‘active components’ of herbal origin should be declared not by the plant name alone but including the details (type of extract etc.).
It is stated that myrrh is an extract of the resin of Commiphora myrrha. When following the definitions as in the European Pharmacopoeia Myrrh is the resin, nothing is extracted. So clarification is required whether actually an extract of myrrh or myrrh itself is used.
The results regarding propolis must be interpreted very carefully as the composition of propolis depends on the area where the insects collected the material and the flowering plants there. Therefore no general conclusion on the effects of propolis is possible without further chemical characterisation.
The discussion and the conclusions contain numerous speculations about the reasons for the effects found in the experiments. The value of the manuscript could be significantly increased when the authors try to find out unique constituents of the plant extracts instead of making reference to e.g. tannins which are present in nearly every plant.
Author Response
Thank you for the review of our manuscript entitled “The effect of toothpastes containing natural extracts on bacterial species of a microcosm biofilm and on enamel caries development”. We appreciate all the suggestions and comments made about this manuscript. Please find below the answers for the reviewers’ comments. The changes were marked up using the “Track changes” in the main document.
Reviewer: 3
Comments and Suggestions for Authors
Details regarding the herbal extracts: Although reference is made to an earlier publication the authors should specify the herbal preparations as precise as possible (kind and strength of the extraction solvent, drug-extract-ratio, type of the extract (liquid extract, dry extract). Without these details the statements regarding the content (e.g. Vochysia tucanorum 10 mg/g) are meaningless.
Answer: The detailed description of the procedure to obtain the extracts was added as requested in lines 91 to 107.
In table 1 the ‘active components’ of herbal origin should be declared not by the plant name alone but including the details (type of extract etc.).
Answer: More information about the type of the extract (plant parts and solvents) were detailed in Table 1.
It is stated that myrrh is an extract of the resin of Commiphora myrrha. When following the definitions as in the European Pharmacopoeia Myrrh is the resin, nothing is extracted. So clarification is required whether actually an extract of myrrh or myrrh itself is used.
Answer: We provided the information presented at the Label of the commercial toothpaste. The manufacture did not provide further information about the plant, we apologise.
The results regarding propolis must be interpreted very carefully as the composition of propolis depends on the area where the insects collected the material and the flowering plants there. Therefore no general conclusion on the effects of propolis is possible without further chemical characterisation.
Answer: Once again, we provided the information presented at the Label of the commercial toothpaste. The manufacturer did not provide further information about the plant extraction even upon request of further information about the active components. This was discussed in lines 303 to 307.
The discussion and the conclusions contain numerous speculations about the reasons for the effects found in the experiments. The value of the manuscript could be significantly increased when the authors try to find out unique constituents of the plant extracts instead of making reference to e.g. tannins which are present in nearly every plant.
Answer: The discussion was revised making reference to more specific compounds present in the extracts in lines 316 to 335.
Reviewer 4 Report
This study investigated the effects of several herbal toothpastes along with two commercial toothpastes on bacterial counts and enamel demineralization. The study found that all the toothpastes could inhibit oral bacteria growth and reduce mineral loss and lesion depth. The manuscript organized well and the data was present properly. I just have several concerns as following:
- Since the saliva samples were collected from different people, the existing bacteria in saliva samples were different (which means the inoculum is different). Whether the author noticed this point and mixed the saliva samples before inoculation? If the author noticed this, they should make it clear.
- For biofilm analysis, the author should wash the specimens 3 times to remove the planktonic cells first. Also, vortex is not strong enough to get biofilm bacteria out of the biofilm, the author should use sonication, put the biofilm samples in a cold water bath then sonicate for 2-3 min to free bacteria out of the biofilm.
- In tables and figures legend, the author said “different letter show significant differences among the …”, please write it with more detail.
Author Response
Thank you for the review of our manuscript entitled “The effect of toothpastes containing natural extracts on bacterial species of a microcosm biofilm and on enamel caries development”. We appreciate all the suggestions and comments made about this manuscript. Please find below the answers for the reviewers’ comments. The changes were marked up using the “Track changes” in the main document.
Reviewer: 4
Comments and Suggestions for Authors
This study investigated the effects of several herbal toothpastes along with two commercial toothpastes on bacterial counts and enamel demineralization. The study found that all the toothpastes could inhibit oral bacteria growth and reduce mineral loss and lesion depth. The manuscript organized well and the data was present properly. I just have several concerns as following:
1. Since the saliva samples were collected from different people, the existing bacteria in saliva samples were different (which means the inoculum is different). Whether the author noticed this point and mixed the saliva samples before inoculation? If the author noticed this, they should make it clear.
Answer: Saliva samples were collected from 10 donors; the saliva samples from 10 persons were mixed (pool), to avoid differences in the biofilm cultivation. We added this information in the text (lines 87-88).
2. For biofilm analysis, the author should wash the specimens 3 times to remove the planktonic cells first. Also, vortex is not strong enough to get biofilm bacteria out of the biofilm, the author should use sonication, put the biofilm samples in a cold water bath then sonicate for 2-3 min to free bacteria out of the biofilm.
Answer: Thank you for the suggestions. We followed the protocol developed by Henne et al. (Henne K, Rheinberg A, Melzer-Krick B, Conrads G. Aciduric microbial taxa including Scardovia wiggsiae and Bifidobacterium spp. in caries and caries free subjects. Anaerobe. 2015;35:60–5; Henne K, Gunesch AP, Walther C, Meyer-Lueckel H, Conrads G, Esteves-Oliveira M. Analysis of bacterial activity in sound and cariogenic biofilm: A pilot in vivo study. Caries Res. 2016;50:480-8.) for the extraction of bacteria from the biofilm formed around caries lesions. There is actually a washing step included to remove loosely attached (probably planktonic) cells (according to Henne et al. 2015) “For washing, a volume of 250 µl sterile bidistilled water was added to the biofilm pellets, vortexed and cells were again pelleted by centrifugation at 10,600 rcf for two minutes, followed by subsequent discard of the supernatant.” We added this information to the M&M section. Thank you very much for this comment.
3. In tables and figures legend, the author said “different letter show significant differences among the …”, please write it with more detail.
Answer: We modified the legends as requested.
Round 2
Reviewer 2 Report
Thank you for the improvement of the manuscript.
I do agree that every discovery of (new) antimicrobial agents started by measuring the inhibitory effect. But we need scientific evidence on how the compounds are exerting these inhibitory effects to get published in the renowned journal Antibiotics. For Antibiotics, this would be a preliminary screening. Please focus on some molecular mechanistic studies.
There is no evidence for the initial biofilm formation on all the tested enamel after the initial incubations. Data should be presented as before treatment, after treatment for better illustration of the study results. They should use two sets for this study. The molecular analysis of biofilm was done only on the fifth day (the end of cultivation). My point was whether there was similar biofilm formation in all conditions before treatment? If there were significant variations among conditions, that could significantly alter the result. pH analyses were done to check the biofilm formation. Is there any prior evidence that only biofilm would change the pH, not the bacterial growth?
Please clearly mention all points with proper citations including the reason for using different concentrations.
The images are from a pre-experiment (placebo control) provided, but quality and size are low. Please provide larger images mentioning the conditions.
Author Response
Thank you for the review of our manuscript entitled “The effect of toothpastes containing natural extracts on bacterial species of a microcosm biofilm and on enamel caries development”. We appreciate all the suggestions and comments made about this manuscript. Please find below the answers for the reviewers’ comments. The changes were marked up using the “Track changes” in the main document.
Reviewer: 2
I do agree that every discovery of (new) antimicrobial agents started by measuring the inhibitory effect. But we need scientific evidence on how the compounds are exerting these inhibitory effects to get published in the renowned journal Antibiotics. For Antibiotics, this would be a preliminary screening. Please focus on some molecular mechanistic studies.
Answer: Thank you for this interesting comment. We agree that this paper is only a preliminary study, as it is the first to investigate not only the antimicrobial potential but also the anticaries effect of these natural plant extracts. Therefore, at the moment, we unfortunately do not have further molecular analysis to explain the mechanisms of action; but this kind of analyses are already planned for future investigations. However, there is evidence in the literature showing some possible mechanisms involved in the antibacterial action of the natural extracts tested. “A previous study showed that several of the compounds identified in propolis are able to inhibit glycosyltransferase - GTF activities and bacterial growth. Apigenin, for example, is a potent inhibitor of GTF activity, and tt-farnesol was found to be an effective antibacterial agent [Koo H, Rosalen PL, Cury JA, Park YK, Bowen WH. Effects of compounds found in propolis on Streptococcus mutans growth and on glucosyltransferase activity. Antimicrob Agents Chemother. 2002;46(5):1302-9.]. On the other hand, there is no information in the literature about a possible mechanism of action of Myrrha on oral bacteria.” In order to give the readers some insights on this topic, this information was added to the discussion of the manuscript (lines 329-334).
“The antimicrobial effects of chamomile flowers may be attributable to its terpenic derivatives, chamazulene, β-bisabolol, and A and B bisabolol oxides. The mechanism of action of terpenes is not fully understood, but is speculated to involve membrane disruption of bacteria by the lipophilic compounds [Mendoza L, Wilkens M, Urzua A. Antimicrobial study of the resinous exudates and of diterpenoids and flavonoids isolated from some Chilean Pseudognaphalium (Asteraceae) J Ethnopharmacol. 1997;58:85–8.]”. We also added the information on discussion section to improve the manuscript. Thank you for your comments.
There is no evidence for the initial biofilm formation on all the tested enamel after the initial incubations. Data should be presented as before treatment, after treatment for better illustration of the study results. They should use two sets for this study. The molecular analysis of biofilm was done only on the fifth day (the end of cultivation). My point was whether there was similar biofilm formation in all conditions before treatment? If there were significant variations among conditions, that could significantly alter the result.
Answer: As we standardized the microorganism source, the volume of human saliva pool applied in each well and all the conditions of biofilm growth, we expect the that the biofilm amount at 24h (the initial condition before treatment) was the same between all groups. It is not possible to analysis the initial biofilm formation (biofilm thickness or weight, number of bacteria, for example) without disturbing the biofilm, which for sure would have had a negative impact on its growth along to the experimental period. So, we do not have this analysis to add in the paper, except the pH values at the bottom of 24h-biofilm. We controlled the biofilm formation every day both by our naked eyes and by using a digital single-lens reflex camera to obtain images of the 24-well microtiter plates during biofilm growth as well as after treatment and at the end of the experiments, to register obvious signs of biofilm formation. To make this point clearer we had included new images in the manuscript (see Figure 2).
pH analyses were done to check the biofilm formation. Is there any prior evidence that only biofilm would change the pH, not the bacterial growth?
Answer: Thank you for this question. The pH analysis was not done to check the biofilm formation, but the impact of the treatments and the presence of sucrose in the pH variation along to the experiment period, as it is related to caries development. However, to answer your question and in according to that, we have shown in two previous studies the correlation between the changes in the biofilm pH (especially in the presence of sucrose) and changes in the relative bacterial/metabolically activity of the biofilm of natural cavitated caries lesions.
It has been demonstrated from one side that the relative genome abundance (more related to bacterial growth) showed no significant correlation to changes in the pH of bacterial cultures containing both caries-associated as well as non-caries associated bacterial species. On the other hand, in the presence of sucrose and for caries-associated species there was a significant correlation between changes in the relative bacterial activity and the pH decrease. Additionally, the rise of ribosomes counts for, for example of L. paracasei, has shown a directly correlation with the sucrose pulse, achieving its maximum coincidently with the most intense pH drop [Walther C, Meyer-Lueckel H, Conrads G, Esteves-Oliveira M, Henne K. Correlation between relative bacterial activity and lactate dehydrogenase gene expression of co-cultures in vitro. Clin Oral Investig. 2019;23:1225–35]. Also, in vivo high ribosomes numbers for L paracasei have been observed in the biofilm around cavitated carious lesions [Henne K, Gunesch AP, Walther C, Meyer-Lueckel H, Conrads G, Esteves-Oliveira M. Analysis of Bacterial Activity in Sound and Cariogenic Biofilm: A Pilot in vivo Study. Caries Res. 2016;50(5):480-488]. Therefore, there is evidence that the pH drop in carious biofilm is strongly related to the sugar metabolism and the activity of enzymes (such as LDH) needed for the sucrose metabolism and, consequently, to acid production and tooth demineralization. To make this correlation clearer we have included this information in the manuscript (lines: 309-315).
Please clearly mention all points with proper citations including the reason for using different concentrations.
Answer: The reason for testing different concentrations were already explained in the last review.
The images are from a pre-experiment (placebo control) provided, but quality and size are low. Please provide larger images mentioning the conditions.
Answer: We added another figure showing the whole microplate containing all groups at 24 hours of biofilm growth (Figure 2).
Reviewer 4 Report
The revised manuscript is good for publication.
Author Response
Thank you very much.